# Tumid Lupus Erythematosus (TLE): A Review of a Rare Variant of Chronic Cutaneous Lupus Erythematosus (cCLE) with Emphasis on Differential Diagnosis

**DOI:** 10.3390/diagnostics14070780

**Published:** 2024-04-08

**Authors:** Maged Daruish, Francesca Ambrogio, Caterina Foti, Alessandra Filosa, Gerardo Cazzato

**Affiliations:** 1Dorset County Hospital NHS Foundation Trust, Dorchester DT1 2JY, UK; maged.daruish@dchft.nhs.uk; 2Section of Dermatology and Venereology, Department of Precision and Regenerative Medicine and Ionian Area (DiMePRe-J), University of Bari “Aldo Moro”, 70124 Bari, Italy; dottambrogiofrancesca@gmail.com (F.A.); caterina.foti@uniba.it (C.F.); 3Pathology Department, “A. Murri” Hospital-ASUR Marche, Aree Vaste n. 4 and 5, 63900 Fermo, Italy; alessandrafilosa@yahoo.it; 4Section of Molecular Pathology, Department of Precision and Regenerative Medicine and Ionian Area (DiMePRe-J), University of Bari “Aldo Moro”, 70124 Bari, Italy

**Keywords:** tumid lupus erythematosus, TLE, systemic lupus erythematosus, cCLE, inflammatory dermatopathology

## Abstract

Tumid lupus erythematosus (TLE) has been the subject of heated debate regarding its correct nosographic classification. The definition of TLE has changed over time, varying according to the different studies performed. In this review, we address the initial definition of TLE, the changes that have taken place in the understanding of TLE, and its placement within the classification of cutaneous lupus erythematosus (CLE), with a focus on clinical, histopathological, immunophenotypical, and differential diagnosis aspects.

## 1. Introduction

Cutaneous lupus erythematosus (CLE) is a chronic autoimmune skin disease characterized by various kinds of cutaneous manifestations that can range from mild skin rashes to more severe lesions and can significantly impact the quality of life of affected individuals [1]. Classically, CLE is divided into the following different forms:(1)Acute cutaneous lupus erythematosus (ACLE), typically presents with a malar rash, which is a butterfly shaped rash across the cheeks and bridge of the nose but that may also involve the scalp, neck, and upper chest [2]. From a histopathological point of view, ACLE presents interface dermatitis, with vacuolar degeneration of basal keratinocytes, often accompanied by lymphocytic infiltration and a perivascular and periadnexal inflammation, with some degrees of dermal edema and, in some cases, signs of leukocytoclastic vasculitis. Furthermore, epidermal changes such as hyperkeratosis, focal parakeratosis, and dyskeratosis can be appreciated. Finally, mucin deposition in the dermis is another potential feature of this form of CLE [3].(2)Subacute cutaneous lupus erythematosus (SCLE) is characterized by nonscarring, psoriasiform, or annular lesions predominantly found on sun-exposed areas such as the upper back, shoulders, extensor surfaces of the arms, and neck and that can also present as widespread erythematous plaques [4]. Histologically, SCLE presents interface dermatitis with basal cell vacuolization and hyperkeratosis with some degree of follicular plugging, which is a common finding in SCLE and contributes to the characteristic scaling and follicular papules seen in SCLE lesions. At the level of the dermis, it is possible to appreciate a dense inflammatory infiltrate around blood vessels (perivascular) and hair follicles (perifollicular) with lymphocytes, histiocytes, and occasionally eosinophils infiltrate these areas, contributing to the inflammatory response. Finally, in some cases of SCLE, there are mucin deposition and dermal changes such as edema [5].(3)Chronic cutaneous lupus erythematosus (CCLE) encompasses several subtypes: discoid lupus erythematosus (DLE) presents with well-defined, scaly, erythematous plaques often with follicular plugging and atrophy and in which the lesions typically occur on the face, scalp, and ears, but can also affect other areas of the body.

The hypertrophic variant of CCLE manifests as thickened, hyperkeratotic plaques, particularly on the scalp and other areas subject to trauma or friction while the mucosal lupus erythematosus can cause oral or nasal ulcerations and can occur in isolation or concurrently with other CLE subtypes.

Histologically, DLE presents epidermis with hyperkeratosis, variable epidermal atrophy alternating with acanthosis, and follicular plugging [6]. The basement membrane is often thickened, and the inflammatory infiltrate has a superficial and deep pattern and frequently involves adnexal structures [7].

Hypertrophic CLE presents hyperkeratosis, acanthosis, and hypergranulosis with inflammatory infiltrate that, although less pronounced than in other variants of cutaneous lupus erythematosus, can be observed in the papillary and reticular dermis. Furthermore, there is almost an absence of interface dermatitis and there is follicular hypertrophy with the absence of atrophy [8].

Tumid lupus erythematosus (TLE) is a rare, chronic relapsing, photosensitive dermatosis of indolent clinical behavior [9,10]. TLE has been traditionally classified under the umbrella definition of chronic cutaneous lupus erythematosus (CCLE) [11]. CCLE includes other cutaneous manifestations that may be seen infrequently in association with systemic lupus erythematosus (SLE) including discoid lupus erythematosus (DLE), lupus panniculitis, and chilblains lupus erythematosus [10,11]. However, since the first use of the term “lupus erythematodes tumidus” in the Berlin Dermatological Society in 1909 [12], the classification of TLE and its genuine relation to lupus erythematosus (LE) has remained disputed in the literature [9,10,13,14,15]. Indeed, TLE was not present in the classification by Gilliam in the 1970s that contemplated three main clinical types: chronic CLE (CCLE), which included discoid lupus erythematosus (DLE) as the most important subtype; subacute CLE (SCLE) and acute CLE (ACLE) [16]. To complicate matters, a spectrum including TLE with lymphocytic infiltrate of Jessner and reticular erythematous mucinosis (REM) has been postulated [17]. Anyway, it is important to remember that TLE, particularly in the European literature, has been neglected mainly because it has not always been considered a separate entity.

In this review article, we discuss the clinicopathological characteristics of TLE with a focus on classification history, related debates, prognosis, and therapeutic approaches.

## 2. Epidemiology

The exact incidence of TLE is unknown but it is less common than DLE [17]. Information regarding the prevalence and incidence of TLE is still lacking, as well as absent specific categorization code in the International Classification of Diseases (ICD), in which it is included in ICD L93.2 (other local lupus erythematosus, such as Chilblain LE and lupus erythematosus profundus) [18]. In contrast to CCLE, which is more common in females, TLE appears to have equal sex incidence or a slight male predilection [9,13]. The age of presentation can be wide but mostly seen in the fourth and fifth decades [9,13,14]. Presentation in children is rare [19]. In 2013, the European Society of Cutaneous Lupus Erythematosus (EUSCLE) provided clinical data from 1002 CLE patients from 13 European countries and Brazil, and only 65 of these were diagnosed with TLE and a further 41 were diagnosed with TLE together with one or more different CLE subtypes, most commonly ACLE or DLE [20].

## 3. Etiopathogenesis

Etiopathogenesis of TLE has not been yet fully elucidated but is believed to involve a complex interplay between genetic, environmental, and immunological factors [19,21,22]. Ultraviolet (UV) radiation is a major triggering factor in most, but not all TLE patients [15,23]. In fact, TLE was found to be the most photosensitive type of cutaneous lupus erythematosus [23]. UV radiation can lead to the induction of keratinocytes apoptosis and exposure of autoantigens to circulating antibodies [24]. Cigarette smoking has been associated with both TLE and DLE [25]. Multiple drugs have been described to induce TLE such as tumor necrosis factor-alpha (TNF-a) inhibitors, angiotensin-converting-2 enzyme inhibitors, and bortezomib [26,27,28,29]. TLE has rarely been reported in association with DLE and SLE [30,31]. Indeed, TLE appears to be the least form of CCLE to be seen concomitantly with SLE. ANA titers are typically low (≤1:160), and higher titers should raise concerns of systemic involvement with LE [31]. Immune dysregulation, however, appears to play an important role in the pathogenesis of TLE. FOXP3+ and CD39+ T-regulatory cells and epidermal Langerhans cells were shown to be decreased in TLE [24,32]. Similar to other types of cutaneous lupus erythematosus, plasmacytoid dendritic cells (pDCs) recruitment is believed to be a major factor in the pathogenesis, with the resultant production of Interferon type I (IFN-I). The latter subsequently leads to the activation of T-lymphocytes and the induction of chemokines and cytokines [22,33].

## 4. Clinical Picture

The characteristic clinical presentation of TLE is that of erythematous edematous urticarial plaques on sun-exposed sites. The lesions may have an annular or arciform configuration. Epidermal changes such as ulceration, scaling, and crusting are typically absent and the lesions tend to heal without secondary sequalae but can recur [21,31,34]. As already mentioned, there is a predilection for sun-exposed areas such as the neckline, shoulders, face, and arms. TLE in “blaschkoid” distribution has been reported [35]. Also, TLE may present with periorbital edema, or scalp involvement similar to alopecia areata (AA). Lesions of TLE persist for days or weeks with the potential to regress spontaneously; however, patients may report recurrence during the summer months [34,35].

Figure 1 represents an example of TLE on the right cheek of a 25-year-old female.

## 5. Histopathology

From a histopathological point of view, there is abundant dermal mucin deposition with a superficial and deep perivascular and peri-adnexal lymphocytic infiltrate and occasional edema in the papillary dermis. Typically, the epidermis and the dermo-epidermal junction are spared with the absence of atrophy, scarring, follicular plugging, and dyspigmentation [36] (Figure 2).

Direct immunofluorescence studies are usually negative (differential diagnosis with “lupus band test” of ACLE/SLE patients. It is important to emphasize that histopathological findings may show variations according to the body area biopsied, the specific location selected on the plaque (central or peripheral), and the time of taking a biopsy (recent or older lesion) are factors that may play a role in the histologic interpretation in TLE, as in many other inflammatory skin conditions [37].

Furthermore, the staining with anti-CD123 (IL-3 receptor alpha-chain) antibody is important to confirm the histological diagnosis of TLE, since clusters of PDCs are quite characteristic for lupus dermatitis. The role of PDCs in the pathogenesis of TLE has been underlined recently and there is robust evidence that the PDCs are important actors in the inflammatory mechanisms behind TLE [1,38] (Figure 3). 

## 6. Classification of TLE

As part of the spectrum of LE, the nosology of TLE has been evolving. Whereas rheumatological classification for LE is based on symptoms and assessment of systemic involvement, from a dermatological perspective, classification has been established in order to categorize the disease mainly according to the morphology of skin lesions [22,39,40].

In the classification systems for dermatological purposes, chilblain lupus, lupus tumidus, and lupus profundus are predominantly classified as cutaneous-limited LE [22,39]. Since the first case described in the early 1900s, Gougerot and Burnier [41] described five patients with similar clinical features, such as erythematous, indurated, non-scarring lesions on the face with minimal surface changes. In the following years, the true incidence of TLE was likely underestimated as the next case reports of TLE were published in the European literature in the 1950s [42,43]. This might have been due to the fact that authors did not consider TLE as a separate entity differing from other variants of cutaneous lupus erythematosus. In 1981, the classification proposed by Gilliam and Sontheimer differentiates LE-associated cutaneous lesions into specific and nonspecific entities based on histology [11]. The specific ones, defined by the presence of dermo-epidermal interface dermatitis, are exclusively seen in LE, with or without systemic disease. They are subdivided into three categories based on clinical characteristics: acute cutaneous lupus erythematosus (ACLE), subacute cutaneous lupus erythematosus (SCLE), and chronic cutaneous lupus erythematosus (CCLE) [9,15,44]. The nonspecific lesions include other cutaneous manifestations associated with SLE. In 2004, the Düsseldorf classification added another subtype, the intermittent CLE (ICLE), which corresponds to tumid LE, previously considered as a variant of CCLE [45].

However, with increasing evidence, interface dermatitis, used as a criterion to define specific CLE lesions, was found to lack specificity, as it may be present in other conditions, such as dermatomyositis, graft-versus-host disease (GVHD), and drug reactions [14,44]. In the last two decades, the extensive works of Kuhn and co-workers on TLE demonstrated several significant differences between TLE and other subtypes of CLE [13,46]. These differences were based on clinical, histological, and laboratory parameters and indicate that TLE should be defined as a separate entity in the classification of CLE. They defined diagnostic criteria for the classification of the disease so that a correct diagnosis requires attention to subtle details, identification of the characteristic signs as well as the course of the disease [47].

Table 1 summarizes the most important differential features between the forms of LE.

## 7. Prognosis

Although TLE is currently considered to be a subtype of CLE, TLE differs from the other subtypes of CLE in that an association with SLE is rare [13,44,48]. Because of this weak association with SLE and a relative lack of serologic abnormalities in patients with TLE, it has been defined as a disease with a benign course [13,34,48]. In a very recent study [44], the authors report the clinical/epidemiological features of 179 patients with TLE of which 15 (8.4%) had ≥1 concurrent diagnosis of lupus subtypes: 5 were preceded by SLE, 6 were preceded by DLE, while SLE subsequently developed in 3; and both chilblain lupus and TLE simultaneously developed in 1.

Long-term remission is observed in some patients. TLE lesions have a favorable prognosis compared to discoid lupus erythematosus or subacute cutaneous lupus erythematosus [13,14,34,48]. Spontaneous resolution of the lesions without residual dyspigmentation or scarring may be noted within days or weeks [34]. Solitary lesions are mostly self-limiting, often without any need for topical or systemic therapy. However, recurrences are frequent with a relatively long disease-free period in between. Risk factors for relapses include sun exposure and smoking [44].

## 8. Therapy

Although singular TLE lesions are frequently self-limiting, there is a high relapse rate [34]. Singular lesions responding quickly to topical therapies may not need any further treatment [49]. Sunscreens, topical corticosteroids, and systemic antimalarials are the most common and most frequently used effective therapeutic measures. Sun protection is recommended in all patients with TLE. Earlier case series reported a high response rate with sun protection and topical corticosteroids, with 19% to 55% of patients requiring subsequent systemic anti-malarials [50,51]. Hydroxychloroquine 200 to 400 mg daily is considered the first-line systemic treatment for TLE. Its response rate varies among studies and may be influenced by dosage [34,50,51]. Second-line treatments include methotrexate 7.5 to 25 mg once weekly, thalidomide 50 to 100 mg daily, and quinacrine. However, quinacrine is not currently commercially available. Thalidomide and quinacrine represented useful alternatives when hydroxychloroquine monotherapy failed. As with other immunomodulators, adverse effects should be monitored periodically. Data regarding the efficacy of systemic agents used as second-line treatment of CLE are lacking in terms of therapy for TLE. Such treatments include methotrexate, and retinoids such as acitretin, dapsone, mycophenolate mofetil, and thalidomide, all of which are preferably used in combination with antimalarials [52]. The treatment of TLE with pulsed dye laser (PDL) has been evaluated in a monocentric prospective study. All ten patients showed clinical improvement. However, relapses were not prevented, and new lesions developed in 50% of the patients. Thus, it is not considered as a treatment option for patients with TLE [34]. Recently, the anti-CD20 monoclonal antibody rituximab was successfully used in treating relapsing TLE lesions [53], but the effectiveness of this approach needs to be fully demonstrated since B-lymphocytes are not recognized as important players in the immunologic pathogenesis of TLE [54]. The elimination of photosensitizing drugs should be also considered, especially in refractory disease [20]. Lastly, TLE patients are advised to join smoking cessation programs because of the evidence-based association of their disease with smoking [25]. Moreover, patients might be prone to vitamin D deficiency by avoiding sun exposure. Hence, evaluation of the vitamin D deficiency with a 25-hydroxyvitamin D level and adequate supplementation with at least 400 IU of cholecalciferol is suggested [52].

## 9. Differential Diagnosis

### 9.1. Jessner–Kanof Infiltrate (Pseudolymphoma)

In terms of differential diagnosis, it is important to emphasize the main clinico-pathological entities from which TLE must be differentiated. The relationship between TLE and the Jessner–Kanof lymphocytic infiltrate has not been always clearly defined. However, since the initial description by Jessner [55], the lymphocytic infiltrate (pseudolymphoma) has been considered a different entity from TLE. Weber F. et al. [56] performed a photobiology study including 10 patients who had Jessner lymphocytic infiltration. The investigation revealed that all the patients experienced a latency period exceeding 48 h prior to the onset of lesions, which is common in all types of CLE. The patients also acquired lesions upon photoprovocation. Examining the biopsies from the lesions showed that the epidermis was unaffected and that the perivascular and periadnexal infiltration was identical to that of TLE. They concluded that there were no appreciable variations between Jessner lymphocytic infiltration and TLE in terms of clinical, pathological, or photobiological aspects. Furthermore, Rémy-Leroux et al. [57] compared 14 cases of TLE with 32 cases of Jessner lymphocytic infiltration. They concluded that Jessner lymphocytic infiltration and TLE are interchangeable after examining the clinical, microscopic, and response characteristics of the two patient groups in the photobiology investigation. The results of those investigations led to the current consensus that Jessner lymphocytic infiltration is not a distinct illness, but rather a subtype of TLE.

### 9.2. Polymorphous Light Eruption (PLE)

Another differential diagnosis is polymorphous light eruption (PLE) is a photodermatosis that can result in a wide range of skin lesions, although these are often monomorphic in a given patient. Skin lesions can be vesicular, pseudovesicular, or can resemble papules or plaques that are difficult to distinguish from TLE. In the latter instance, determining the differential diagnosis requires an understanding of the specific variations in the lesions’ clinical course. Unlike TLE, polymorphous light eruption lesions appear shortly after exposure to sunlight and heal on their own in a few days if no additional exposure occurs. Additionally, if exposure is prolonged, a highly distinctive tolerance phenomenon develops, and the recurrences become less severe [9,58]. Another important differentiating feature is the CD123 immunostaining, highlighting clusters of PDC in TLE but few or no cells in cases of PLE.

### 9.3. Reticular Erythematous Mucinosis (REM)

Steigleder G.K. et al. [59] presented cases in 1974 that were indistinguishable from those of TLE but termed them “Reticular erythematous mucinosis” and many authors since have published their observations under the name of REM [60,61,62,63]. The majority of patients with REM are young women, which usually manifests as papular erythema or a reticulated macular rash. A periadnexal and perivascular lymphocytic infiltration associated with interstitial mucin is found histologically [58]. In addition, most patients experience significant photosensitivity. This has led some authors to believe that reticular erythematous mucinosis is a form of CCLE or TLE [9].

Interestingly, immunoglobulin M deposits have been reported on the basement membrane in REM [64]. However, no published comparison studies between the two REM and TLE regarding immunofluorescence have been conducted to provide confirmation of this as a differentiating factor.

### 9.4. Granuloma Faciale (GF)

Another clinically important differential diagnosis is granuloma faciale (GF), which typically presents as an asymptomatic violaceous nodule or plaque on the face. Histologically, a *Grenz zone* is the characteristic histologic finding that distinguishes it from TLE; a thin zone of the uninvolved papillary dermis that divides the dermal inflammatory infiltrate constituted by lymphocytes, histiocytes, plasma cells, neutrophils, and eosinophils [65].

### 9.5. Cutaneous Manifestations of B-Chronic Lymphocytic Leukemia (B-CLL)

Differential diagnosis with B-CLL is of paramount importance and the co-expression of CD20, CD5, and CD23 by neoplastic cells of B-CLL as well as the detection of a monoclonal rearrangement of the Ig genes in contrast to the polyclonality of T lymphocytes admixed with CD123+ PDC in TLE, allows for reaching the correct diagnosis [38].

## 10. Associations Rarely Reported

TLE has rarely been reported in association with uncommon histopathological features that deserve specific mention in this review. In 2022, Georgiadou N. et al. [66] reported a case of an 85-year-old patient who presented with four indurated erythematous plaques on her face and upper chest. A 4 mm punch biopsy of one of these lesions showed in addition to the histological features compatible with lupus dermatitis, haemophagocytosis (mainly erythrophagocytosis, lymphophagocytosis, and cellular debris). Another paper [67] described the occurrence of neutrophilic granulocytes in the context of the typical dermal lymphocytic infiltrate of CLE. Furthermore, Boggio F et al. [68], in 2018, reported the largest published series in the literature of 21 cases of cutaneous haemophagocytosis in patients who had underlying pathological conditions not ascribable to cutaneous T-cell lymphoma and cutaneous Rosai–Dorfman diseases (cRDD) and presented, among other cases, two patients with CLE who had histological evidence of inflammatory lymphocytic infiltrate with concomitant haemophagocytosis, mostly represented by neutrophil phagocytosis. As detailed previously [34,66], although there is still no unifying theory explaining the pathogenesis of TLE, it is important to remember that LE is a complex pathology, with a broad spectrum of clinico-pathological manifestations reflecting the two main components at play in its etiology: (1) dysregulation of cell-mediated immunity and (2) immunocomplex deposition pathology. It is therefore plausible to hypothesize that the impairment of this labile balance between the innate and adaptive immune system is also responsible for the pathogenesis of TLE, with the possibility of the development of cutaneous haemophagocytosis, which can also be observed in the course of autoimmune diseases, infections, and malignant neoplasms, among others [69].

## 11. The Larger Series of TLE Reported in Literature

The largest series of patients with TLE reported in the literature so far was published by Magana M. et al. in 2022 [70]. These authors reported data on 20 patients (11 men and 9 women, with an average age of 43.5 years) who had been diagnosed in the previous 16 years with histological activity. All reported patients presented with erythematous, non-scarring urticarial-like plaques, of which only eight were in the head region, eight at the trunk/limbs level; head and trunk/limbs in two cases and topographic data for two patients were not available. Only one patient out of the twenty reported had developed SLE. Histologically, the biopsies showed the classic features of TLE and, interestingly, the differential count of CD123+ PDCs carried out on 10/20 patients with TLE was much higher compared to five cases of DLE and five cases of normal skin, suggesting more robustly that this marker plays a role of some importance in the differential diagnosis of TLE compared to its histological mimics.

## 12. Conclusions

TLE is a type of CLE that differs from the traditional types because of its unique clinical presentation. While the initial accounts of the histological characteristics of TLE emphasized the lack of distinct epidermal changes typical of lupus, certain cases do exhibit these changes, but they tend to be mild or moderate. In order to prevent cases of TLE in which damaged epidermis is found to be mistakenly identified as CCLE or SCLE, the sporadic occurrence of epidermal changes in TLE should be considered in clinical practice, although without crust and/or scarring. It is important to say that today, since the pioneering work by Kuhn and more recently by Magana, TLE can be considered a highly photosensitive disease, with characteristic and peculiar clinical, histopathological, and immunophenotypic features that confirm it as a distinct type of CLE.

Future papers with more cases are needed to improve knowledge of this entity that has been neglected for some time in the literature and has come to the attention of dermatologists and dermatopathologists in recent years.

## Figures and Tables

**Figure 1 diagnostics-14-00780-f001:**
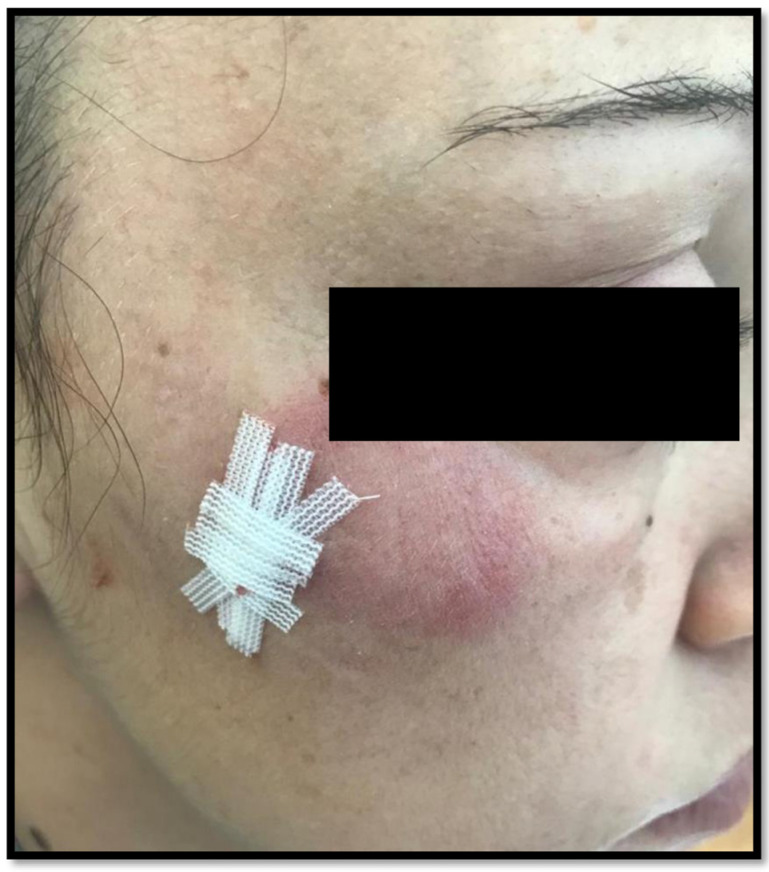
Clinical example of a case with erythematous, edematous, urticarial plaque on the right cheek of a 25-year-old female.

**Figure 2 diagnostics-14-00780-f002:**
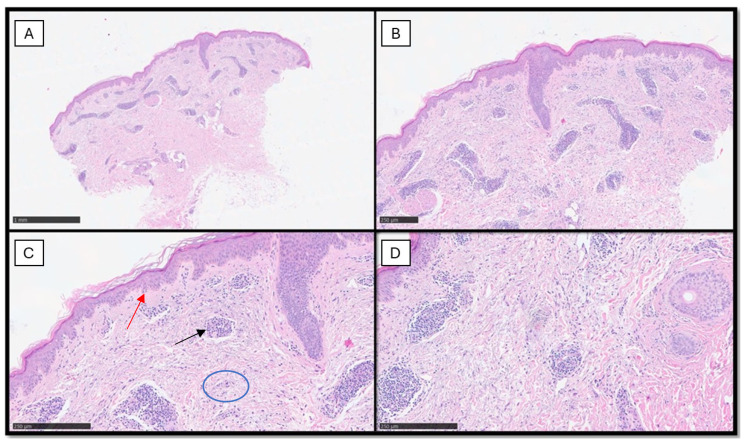
(**A**) Scanning magnification of an incisional biopsy of the same case: note the diffuse perivascular infiltration of inflammatory cells appreciable on low power (hematoxylin–eosin, original magnification 2×). (**B**) Higher magnification shows the perivascular lymphoid inflammatory infiltrate with uninvolved dermo-epidermal junction (hematoxylin–eosin, original magnification 4×). (**C**) The perivascular lympho-monocytic infiltrate (black arrow) without involvement of the dermo-epidermal junction (an example indicated by red arrow): note the presence of dermal mucin (blue circle) (hematoxylin–eosin, original magnification 10×). (**D**) Higher magnification of the previous pictures (hematoxylin–eosin, original magnification 20×).

**Figure 3 diagnostics-14-00780-f003:**
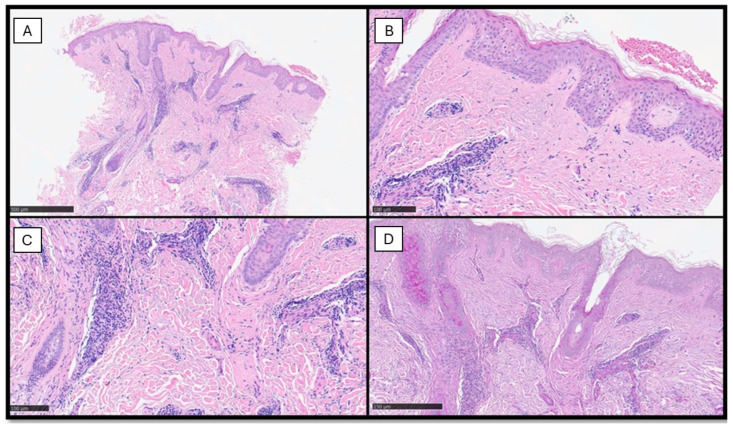
(**A**) Scanning view of an incisional biopsy of another TLE case: note the diffuse perivascular infiltration of inflammatory cells appreciable in this magnification (hematoxylin–eosin, original magnification 2×). (**B**) Higher shows complete absence of inflammation along the dermal–epidermal junction (hematoxylin–eosin, original magnification 10×). (**C**) Details of the previous images (hematoxylin–eosin, original magnification 20×). (**D**) PAS staining showing mild thickening of the basal membrane zone (PAS histochemical staining, original magnification 10×).

**Table 1 diagnostics-14-00780-t001:** Most important clinical, topographical and histological features of the differenti kinds of LE.

Type of LE	Clinical Features	Topography	Histopathological Findings
TLE	Annular, erythematosus, edematous, urticarial plaques	Sun-exposed areas (face, chest, upper back, upper extremities)	Abundant dermal mucin deposition, superficial and deep perivascular and periadnexal lymphocytic infiltrate, with occasional edema in the papillary dermis
DLE	Erythematosus, hyperkeratotic lesions with frequent atrophic scarring	Head and neck, extensor aspects of the arms	Hyperkeratosis, follicular plugging, vacuolar degeneration, thickening of the dermo-epidermal junction
SCLE	Annular, polycyclic lesions with scaly surface, nonscarring, erythematosus	Sun-exposed areas	Vacuolar degeneration with a superficial perivascular and interstitial lymphocytic infiltrate

Legend. TLE: tumid lupus erythematosus; DLE: discoid lupus erythematosus; SCLE: subacute lupus erythematosus.

## Data Availability

Not applicable.

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
