# Peer review of "Tumid Lupus Erythematosus (TLE): A Review of a Rare Variant of Chronic Cutaneous Lupus Erythematosus (cCLE) with Emphasis on Differential Diagnosis"

_diagnostics, 2024, doi:10.3390/diagnostics14070780_

Round 1

Reviewer 1 Report

Comments and Suggestions for Authors

This is a very well written and comprehensive review on a very rare condition in dermatology and dermatopathology. I congratulate the authors on this excellent manuscript.

One small suggestion in the differential diagnosis part. Granuloma faciale might look similar to TLE clinically, but histologically the morphologic characteristics are different (neutrophils and eosinophils in addition to lymphocytes, usually not perivascular distribution of the inflammatory cells, and the presence of leukocytoclastic vasculitis).

Comments on the Quality of English Language

Minor grammar, particularly commas

Author Response

Reviewer n'1

This is a very well written and comprehensive review on a very rare condition in dermatology and dermatopathology. I congratulate the authors on this excellent manuscript.

One small suggestion in the differential diagnosis part. Granuloma faciale might look similar to TLE clinically, but histologically the morphologic characteristics are different (neutrophils and eosinophils in addition to lymphocytes, usually not perivascular distribution of the inflammatory cells, and the presence of leukocytoclastic vasculitis).

Answer n'1: Dear Reviewer n'1, thank you very much for this useful comment. We have added this aspect regarding the clinical overlapping of TLE with GF and not from an histopathological point of view. Thanks again.

Reviewer 2 Report

Comments and Suggestions for Authors

Although it is an interesting topic, and provides useful information

1.       The diagnosis of TLE is usually difficult, on the other hand, inflammatory cell infiltration is seen in the area of lupus diseases, so it is difficult to definitively diagnose TLE with pathology. It is mentioned here that monoclonal antibody drugs cause exacerbation of the disease, while many monoclonal drugs such as adalimumab reduce cell signaling and, as a result, reduce the migration of inflammatory cells. So there is still a long way to diagnose and specific treatment of TLE

2.       The manuscript is well presented, but it is better to record the results of the comparison of lupus types in a table.

3.       The references should be corrected, for example, 17, 37 and 60.

4.       Reference 43 and 15 are repeated.

5.       In Figure 2: A, B... should be specified. Figure 1, 2, and 3 are the same, as well as the magnification of 2 and 3

6.       It should also be corrected in Figure 3.

7.       Correct keywords

8.       The font and size should be the same in all the text.

9.       This manuscript is available in preprint form on the site (https://www.preprints.org/manuscript/202403.0947/v1)

Author Response

Reviewer n’2: 1.       The diagnosis of TLE is usually difficult, on the other hand, inflammatory cell infiltration is seen in the area of lupus diseases, so it is difficult to definitively diagnose TLE with pathology. It is mentioned here that monoclonal antibody drugs cause exacerbation of the disease, while many monoclonal drugs such as adalimumab reduce cell signaling and, as a result, reduce the migration of inflammatory cells. So there is still a long way to diagnose and specific treatment of TLE.

Answer n’1: Thank you very much for this useful comment.

Reviewer n’2: 2.       The manuscript is well presented, but it is better to record the results of the comparison of lupus types in a table.

Answer n’2: Dear Reviewer n’2, OK. Thank you. Done.

Reviewer n’2: 3.       The references should be corrected, for example, 17, 37 and 60.

Answer n’3: Thank you very much. We corrected them.

Reviewer n’2: 4.       Reference 43 and 15 are repeated.

Answer n’4: Thank you. We replaced the reference 43 with a new reference. Thank you.

Reviewer n’2: 5.       In Figure 2: A, B... should be specified. Figure 1, 2, and 3 are the same, as well as the magnification of 2 and 3.

Answer n’5: Done. Thank you.

Reviewer n’2: 6.       It should also be corrected in Figure 3.

Answer n’6: Done.

Reviewer n’2: 7.       Correct keywords.

Answer n’7: Done. Thank you.

Reviewer n’2: 8.       The font and size should be the same in all the text.

Answer n’8: Done.

Reviewer n’2: 9.       This manuscript is available in preprint form on the site (https://www.preprints.org/manuscript/202403.0947/v1).

Answer n’9: Yes. Thank you.

Reviewer 3 Report

Comments and Suggestions for Authors

The manuscript presents a comprehensive review of a rare, but important cutaneous disease:  Tumid Lupus Erythematosus (TLE).  The authors have succeeded in presenting this form of lupus from every aspect.

Only few questions are needed to be answered.

In prognosis.

-Provide epidemiological data about associating with SEL and the possible explanation for why the association is low.  

- What are the risk factors for relapses?

In differential diagnosis

-        Are there data about the delay in diagnosis of TLE, because of such broad differential diagnosis?

Author Response

Reviewer n’3: The manuscript presents a comprehensive review of a rare, but important cutaneous disease:  Tumid Lupus Erythematosus (TLE).  The authors have succeeded in presenting this form of lupus from every aspect.

Answer n’1: Thank you very much dear Reviewer n’3 for this beautiful comment. Thanks a lot.

Reviewer n’3: -Provide epidemiological data about associating with SEL and the possible explanation for why the association is low. 

Answer n’2: Thank you very much. We have added the clinical/epidemiological informations regarding a very recent paper.

Reviewer n’3: - What are the risk factors for relapses?

Answer n’3: Thank you for this question. We have added some informations about this topic. Thank you.

Reviewer n’3:  Are there data about the delay in diagnosis of TLE, because of such broad differential diagnosis?

Answer n’4: Thank you for this interesting question. Analyzing the studies it seems that are not present clear data about the delay time for the diagnosis.